# Universal relations in Bose gases with power-law interactions in two and three dimensions

Nina S. Voronova[1,2*], Elena V. Skirdova[1], I. L. Kurbakov[3] and Grigory E. Astrakharchik[4†]

**1** National Research Nuclear University MEPhI (Moscow Engineering Physics Institute), Moscow, Russia
**2** Russian Quantum Center, Skolkovo Innovation Center, Moscow, Russia
**3** Institute for Spectroscopy of the Russian Academy of Sciences, Troitsk, Moscow, Russia
**4** Departament de Física, Universitat Politècnica de Catalunya, Barcelona, Spain

⋆ nsvoronova@mephi.ru ,   † grigori.astrakharchik@upc.edu

## Abstract

We derive universal relations between thermodynamic quantities and the pair distribution function for bosons interacting via power-law interactions in three and two dimensions. We find beyond mean-field expressions for Tan's contact and establish new connections between the chemical potential and potential energy per particle, characteristic of systems with finite-range interactions. The obtained results are relevant for dipoles, quadrupoles, Rydberg atoms, excitons, van der Waals atoms, and other systems in the dilute limit.

July 8, 2024

# 1 Introduction

Ultradilute atoms provide an excellent platform for high-precision experiments due to their purity and the high degree of controllability of both interactions and external potentials. Quantum gases have enabled experimental observation of a great variety of phenomena such as the BCS-BEC crossover [1], superfluid — Mott Insulator phase transitions [2], formation of solitons [3] among others, with an unprecedented degree of control not achievable in conventional condensed matter systems. At the same time, the majority of the observed phenomena have been previously present in other, perhaps not such clean, systems, e.g. for solid-state quasiparticles such as excitons, exciton-polaritons, etc. [4–6]. In that sense, the concept of the *contact* stands out as being genuine to ultracold atoms with short-range interactions [7,8]. It provides a powerful set of relations between thermodynamic quantities and the ultraviolet behavior of the correlation functions (pair distribution function, momentum distribution, one-body density matrix, spectral functions, etc).

The search for universal relations for both fermionic and bosonic ultracold gases began since the coefficient in $1/p^4$ behavior of the high–$p$ tails of the momentum distribution for one-dimensional bosons was related to the equation of state by Maxim Olshanii [7] in 2003. It was soon realized that similar relations hold for fermions [9]. The research has culminated in an influential series of works by Shina Tan where the most celebrated relations in terms of 'the Tan's contact' $\mathcal{C}$ [8,10,11] or a single dimensionless function of the $s$-wave scattering length $a$ and temperature [12] have been derived, expressing important thermodynamic observables in terms of the same parameter. The contact and the corresponding Tan's relations are exact for zero-range contact interactions and have been theoretically studied [13–27] and experimentally observed in three-dimensional [28–32] and planar [33] Bose gases as well as in unitary Fermi gases and across BCS-BEC crossover [34–38]. The power of the Tan's contact is that it relates the correlation functions to the equation of state. The equation of states have a long history of studies and are known in spin-1/2 fermions in three- (3D) [39,40], two- (2D) [41], and one-dimensional [42] (1D) geometry.

It is a notable feature of ultracold atoms that the range of the interaction potential is significantly smaller than the mean interparticle distance. Under such conditions, the specific form of the interaction potential is no longer important, and the interaction potential $V(r)$ can be characterized by a single parameter, namely the $s$-wave scattering length $a$. In the dilute regime, where the atomic density $n$ is low, many system properties are *universal*, i.e. they depend only on the gas parameter $na^3$. In the universal regime, the range of the potential and the effective range in the scattering problem are not important. Since Tan's relations are exact for a contact pseudopotential, $V(r) = g\,\delta(\mathbf{r})$, they are applicable in the universal regime, where the coupling constant $g$ alone is sufficient to describe the interaction potential. However, there has been significant experimental progress in realizing ultracold systems with finite-range potentials, such as dipolar gases [43], Rydberg atoms [44], atom-ion systems [45] where the original description in terms of Tan's contact might be no longer applicable. Thus, a generalization of universal relations to finite-range interactions is an open question addressed in the present work. Here, we demonstrate that zero-temperature Bose gases with finite-range power-law interactions can be treated outside of the universal regime, i.e. not by replacing the real interaction potential with the contact pseudopotential.

The contact $\mathcal{C}$ is related to the ultraviolet behavior of the many-body wave function and, according to the adiabatic sweep theorem [11], emerges as a thermodynamic parameter characterizing the variation of the free energy $\mathcal{F}$ with respect to the coupling constant $g$. The physical reason for the existence of the contact is that for zero-range pseudopotential $V(r) = g\,\delta(\mathbf{r})$, the order of the thermal average $\langle\cdots\rangle$ and the differentiation of the Hamiltonian $H$ with respect to $g$ can be interchanged. This yields either the derivative of a thermodynamic quantity,

i.e., the free energy $d\langle H\rangle/dg = d\mathcal{F}/dg$, or the local value of the density-density correlation function $\langle dH/dg\rangle \propto \langle g\delta(\mathbf{r})\rangle \propto \langle \delta(\mathbf{r})\rangle \propto g_2(0)$. In this way, the short-range behavior of correlation functions is related to the derivatives of the equation of state with respect to $\mathbf{g}$ (for an extensive review, see Refs. [46, 47]). However, this conceptually simple reasoning is not applicable to systems with finite-range interactions. To generalize the concept of the contact to dipolar interactions, Hofmann and Zwerger [48] introduce two contacts: the traditional Tan's contact $\mathcal{C}$ and a new dipolar contact $\mathcal{D}$ in 2D systems with short-range and dipolar interactions.

We consider a very general model of Bose gases interacting with an arbitrary power-law potential

$$V(r) = \frac{Q}{r^\alpha}, \tag{1}$$

where $Q$ denotes the amplitude and $\alpha$ the exponent of the two-body interaction potential. Additionally, we assume that $V(r)$ is integrable at infinity, i.e. $\alpha > 2$ for two dimensions and $\alpha > 3$ in 3D which ensures that at the many-body level, the interaction (1) decays sufficiently fast to give rise to proper thermodynamics with a finite value of potential energy per particle in the macroscopic limit. In the strict sense we consider short-range potentials, although it is common in the cold-gases community to refer to potentials (1) as being long-range ones, in order to differentiate them from short-range potentials. Our aim is to calculate the Tan's contact $\mathcal{C}$ for arbitrary exponent $\alpha$ and derive universal relations without relying on the adiabatic sweep theorem.

In the following we consider $N$ bosons with coordinates $\mathbf{r}_1, \ldots, \mathbf{r}_N$. When the relative distance between any $i$-th and $j$-th particles is small compared to the mean interparticle separation, the influence of all other particles is reduced and the many-body wave function $\Psi(\mathbf{r}_1, \ldots, \mathbf{r}_i, \ldots, \mathbf{r}_j, \ldots, \mathbf{r}_N)$ can be approximately factorized as

$$\lim_{\mathbf{r}_i \to \mathbf{r}_j} \Psi = \psi(r)\tilde{\Psi}(\mathbf{r}_1, \ldots, \mathbf{r}_{i-1}, \mathbf{r}_{i+1} \ldots, \mathbf{r}_{j-1}, \mathbf{r}_{j+1}, \ldots, \mathbf{r}_N), \tag{2}$$

where $\psi(r)$ is the solution for the two-body scattering problem of the relative motion, $\mathbf{r} = \mathbf{r}_i - \mathbf{r}_j$. The scale at which two adjacent particles "forget" about the rest of the gas, is defined by $0 < r \ll \xi$, where $\xi$ is the healing length of the gas (to verify this, we have performed variational Monte-Carlo simulations that are presented in Appendix A). On the other hand, on large scales ($r \gg a$, or for $a \to 0$), one can describe the collective behavior of the $N$-particle Bose gas and determine the pair distribution function from the Bogoliubov theory [49] at $T = 0$. Note that in this reasoning in Eq. (2) one solves the scattering problem on the full interaction potential (1), i.e. without introducing the zero-range approximation, hence there is no requirement $r \gg a$ for the pair distribution function obtained from the solution of the two-body relative motion Schrödinger equation for $\psi(r)$. In the region where these two scales overlap, namely $a \ll r \ll \xi$, two solutions can be matched. Doing so allows us to obtain the Tan's contact $\mathcal{C}$ with a higher accuracy as compared to the result derived from the adiabatic sweep theorem (since the short-range potential properties are not neglected).

Additionally, since the considered interactions possess a finite range, within our theory we are able to express the potential energy in terms of the contact $\mathcal{C}$ and, using the beyond-mean-field equations of state, derive the thermodynamic universal relation between the potential energy per particle and the chemical potential of the gas at $T = 0$.

## 2   Two dimensions

### 2.1   Two-body scattering problem and universal relation for potential energy

To describe correctly the short scales we solve the two-body $s$-wave scattering problem, given by the zero-energy Schrödinger equation of the particle relative motion in power-law potential (1)

$$\left[ -\frac{\hbar^2}{m}\frac{1}{r}\frac{\partial}{\partial r}\left( r\frac{\partial}{\partial r} \right) + \frac{Q}{r^\alpha} \right]\psi(r) = 0, \tag{3}$$

where $m/2$ is the reduced mass of two particles of mass $m$. The angular dependence in the 2D Laplacian and the wave function is omitted as the lowest energy solution is symmetric. Our aim is to calculate the pair correlation function $g_2(\mathbf{r}_i, \mathbf{r}_j)$.

For making the derivation simpler, it is convenient to introduce the dimensionless variables $\rho = r/r_0$ by using as a unit of length the characteristic length associated with the interaction potential, $r_0 = (mQ/\hbar^2)^\nu$ with $\nu = 1/(\alpha - 2)$. By doing so, Eq. (3) is reduced to

$$\psi''(\rho) + \frac{1}{\rho}\psi'(\rho) - \frac{1}{\rho^{2+1/\nu}}\psi(\rho) = 0, \tag{4}$$

which with another substitution $z = 2\nu/\rho^{1/(2\nu)}$ becomes the equation for cylinder functions, with the general solution $\psi = C_1 I_0(z) + C_2 K_0(z)$, where $I_0(z)$ and $K_0(z)$ are modified Bessel functions. We set $C_1 = 0$ to eliminate the divergent term $I_0(z \to \infty) \to \infty$. We then use the asymptotic form $K_0(z \to 0) \approx \ln(2e^{-\gamma_E}/z)$, where $\gamma_E \approx 0.577216$ is the Euler's constant, to match the obtained solution with the logarithmic large-distance behavior of the wave function $\lim_{r\to\infty}\psi(r) = \ln(r/a)$, which is a solution to the free particle equation $\Delta\psi(r) = 0$ in two dimensions. Thus we define $C_2 = 2\nu$ and obtain the following expression for the $s$-wave scattering length for scattering on potential (1):

$$a = r_0 (\nu e^{\gamma_E})^{2\nu} = \left[ \frac{mQe^{2\gamma_E}}{\hbar^2(\alpha - 2)^2} \right]^{1/(\alpha-2)} \tag{5}$$

for any $\alpha > 2$. Note that for purely dipolar interaction, $V_d(r) = d^2/r^3$, for dipoles oriented perpendicularly to the plane of motion, we recover the result $a_d = e^{2\gamma_E}md^2/\hbar^2$ characteristic for 2D dipole-dipole scattering [50]. The scattering solution for potential (1) takes the form

$$\psi(r) = 2\nu K_0\left[ 2\nu\left( \frac{r_0}{r} \right)^{1/(2\nu)} \right] \quad \text{with} \quad \nu = \frac{1}{\alpha - 2}. \tag{6}$$

Characteristic examples of the scattering solution (6) and the scattering length (5) are shown in Figs. 1a and b (inset), respectively. The wave function is significantly suppressed for $r \lesssim r_0$ in the case of the steep potentials ($\alpha > 3$), such as those corresponding to quadrupoles, Rydberg atoms with a van der Waals long-range tail, and excitons described by the Lennard-Jones potential. The extent of this suppressed zone (i.e. an effective "hard-core" radius) increases with the steepness of the potential (i.e., with larger values of $\alpha$). Dipolar interactions, $\alpha = 3$, rather have a "soft core" as the particles are allowed to approach each other more closely. As pointed out in Ref. [48], the effective interaction of dipoles when truncated to motion in the plane needs to contain the short-distance cutoff that ensures the existence of a proper zero-range scaling limit. Our description does not require such a cutoff as we take the finite range of interactions into account.

The pair distribution function $g_2(\mathbf{r}_i, \mathbf{r}_j)$ in a homogeneous system depends only on the relative distance $r = |\mathbf{r}_i - \mathbf{r}_j|$ and, according to the short-distance behavior the solution (2), it

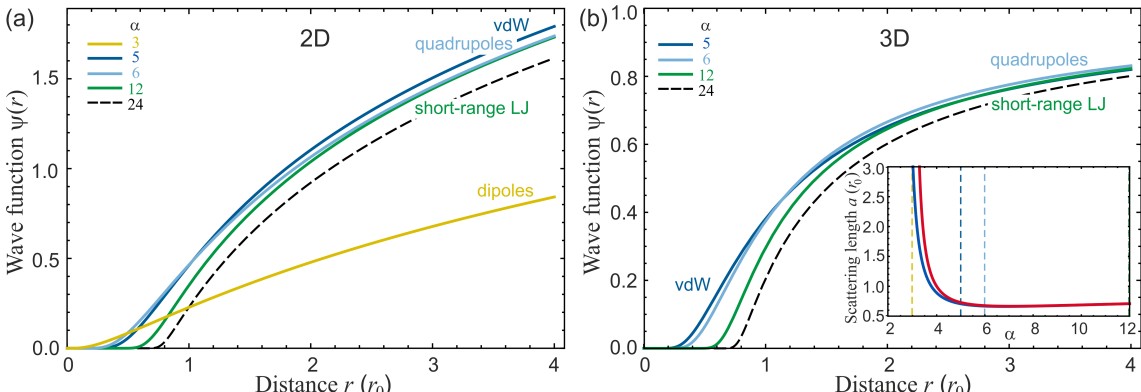

Figure 1: Scattering solution $\psi(r)$ in units of the characteristic length $r_0$ associated with interaction potentials for (a) two-dimensional scattering problem (6) for different interaction exponents, as marked: $\alpha = 3$ (dipole-dipole interaction), $5$ (quadrupole interaction), $6$ (van der Waals interaction), $12$ (Lennard-Jones short-range asymptotic). The black dashed line shows the wave function for $\alpha = 24$. (b) three-dimensional scattering problem (26). Note that while in 2D the solution exhibits logarithmic divergence at large distances, in 3D it is finite and approaches unity at $r \gg a$ as $1 - a/r$. Inset: the scattering length versus $\alpha$ for 2D (blue line, Eq. (5)) and 3D (red line, Eq. (25)) cases. Vertical dashed lines show guides for the eye for the powers used in the main panels.

can be universally defined via the two-body wave function (6) as

$$g_2(r) = \frac{\mathcal{C}}{4\pi^2}\psi^2(r), \tag{7}$$

where $\mathcal{C}$ is the Tan's contact [47]. The homogeneity of the system implies $\lim\limits_{r\to\infty} g_2(r) = n^2$, where $n = N/S$ is the 2D particle density with $S$ being the area of the system. In our convention, the two-body scattering solution $\psi(r)$ is dimensionless, while the Tan's contact $\mathcal{C}$ and the pair distribution function $g_2(r)$ have units of $n^2$. Since we consider the amplitude $Q$ of the interaction potential fixed, no other contact value arises in our description from the change of the energy with respect to variation of $Q$ (cf. Ref. [48] where the variation of the dipole length led to the appearance of a specific relation with a dipolar contact).

Since relation (7) between the scattering solution of the two-body problem and the pair correlation function holds in many-body system but only at short distances, $r \ll \xi$, where $\xi = \sqrt{\hbar^2/(2mgn)}$ is the healing length and $g$ is the coupling constant[1]. Thus this result, due to the integrability of the interaction potential (1) at infinity, can be used to define the potential energy density of the Bose gas in the limit of small densities,

$$u_{2D} \equiv \frac{U}{S} = \frac{1}{2}\int \frac{Q}{r^\alpha}g_2(r)d^2\mathbf{r} = \frac{\mathcal{C}}{4\pi}\int_0^\infty \frac{Q}{r^\alpha}\psi^2(r)r\,dr, \tag{8}$$

where $U$ is the mean potential energy and the scattering solution $\psi(r)$ is given by (6). Integration yields:

$$u_{2D} = \frac{\hbar^2\mathcal{C}}{4\pi m(\alpha - 2)}, \tag{9}$$

---

[1]When any two particles come close to each other, they are weakly affected by the rest of the system and at small distances the pair distribution function $g_2(r)$ can be defined from the scattering problem. As such, if there are just two particles in the box, one has $g_2(r) = 2\psi^2(r)$, where $\psi(r)$ is the normalized wave function of the relative motion. If the box contains $N$ particles, the behavior $g_2(r) \propto \psi^2(r)$ at small $r$ still holds, while the normalization of $\psi(r)$ changes, as the scattering problem is a valid approximation only at distances much smaller than the healing length (see Appendix A). The corresponding proportionality coefficient is $\mathcal{C}/4\pi^2$.

which results in a universal relation between the potential energy density and Tan's contact $\mathcal{C}$ valid for power-law interactions with $\alpha > 2$ in two dimensions and at any temperature (as no assumption about the temperature has been made up to this point).

Finally, the validity of Eq. (7) in the dilute limit allows us to match this solution, obtained from the two-body problem and valid at $0 < r \ll \xi$, with $g_2(r)$ of the full system calculated within the Bogoliubov theory [49] and valid at $r \gg a$. The range of validity of this approach is $a \ll r \ll \xi$. For such distances, we expand the two-body pair distribution function (7) with the substitution of Eq. (6) in the limit $a \to 0$ as

$$\frac{g_2(r)}{n^2} = \mathcal{C}\frac{\ln^2(\varkappa\xi^2/a^2)}{(4\pi n)^2}\left[1 - 2\frac{\ln(\varkappa\xi^2/r^2)}{\ln(\varkappa\xi^2/a^2)} + \mathcal{O}\left(\ln^{-2}\frac{\varkappa\xi^2}{a^2}\right)\right], \tag{10}$$

where we used $\ln(r/a) = [\ln(\varkappa\xi^2/a^2) - \ln(\varkappa\xi^2/r^2)]/2$ and $\varkappa$ is a constant of order 1.

## 2.2 Pair correlation function of the Bose gas

The pair correlation function $g_2(r)$ can be determined from the Bogoliubov theory [49]. In the second quantization approach, the pair correlation function can be expressed in terms of the bosonic field operators according to $g_2(\mathbf{r}, \mathbf{r}') = \langle\hat{\Psi}^\dagger(\mathbf{r})\hat{\Psi}^\dagger(\mathbf{r}')\hat{\Psi}(\mathbf{r}')\hat{\Psi}(\mathbf{r})\rangle$, where the averaging is performed over the ground state of the system at $T = 0$. In the homogeneous case, separating the condensate mode $\mathbf{p} = 0$ in the field operator and performing the Bogoliubov transformation for the non-condensed part

$$\hat{\Psi}(\mathbf{r}) = \sqrt{n_0} + \frac{1}{\sqrt{S}}\sum_{\mathbf{p}\neq 0} e^{\frac{i\mathbf{p}\cdot\mathbf{r}}{\hbar}}\left(u_{\mathbf{p}}\,\hat{\alpha}_{\mathbf{p}} - v_{-\mathbf{p}}^*\,\hat{\alpha}_{-\mathbf{p}}^+\right), \tag{11}$$

provides for the pair correlation function

$$g_2(r) = n_0^2 + \frac{n_0}{S}\left[\sum_{\mathbf{p}\neq 0} e^{-\frac{i\mathbf{p}\cdot\mathbf{r}}{\hbar}}\left(|u_{\mathbf{p}} - v_{\mathbf{p}}|^2 - 1\right) + 2\sum_{\mathbf{p}\neq 0}|v_{\mathbf{p}}|^2\right]. \tag{12}$$

In Eqs. (11-12), $n_0$ is the condensate density, and $\hat{\alpha}$, $\hat{\alpha}^\dagger$ denote the Bogoliubov quasiparticle operators obeying the bosonic commutation algebra. The Bogoliubov amplitudes $u_{\mathbf{p}}$, $v_{\mathbf{p}}$ are given by

$$\{u_{\mathbf{p}}^2, v_{\mathbf{p}}^2\} = \frac{1}{2}\left(\sqrt{1 + \frac{(gn)^2}{E_p^2}} \pm 1\right), \tag{13}$$

with $E_p = \sqrt{p^2/2m(p^2/2m + 2gn)}$ being the Bogoliubov excitation spectrum [51]. Using $(1/S)\sum_{\mathbf{p}\neq 0}|v_{\mathbf{p}}|^2 = n - n_0$ (at $T = 0$) and changing from the summation to integration over the momenta, with the substitution of (13) we get from Eq. (12)

$$\frac{g_2(r)}{n^2} = 1 + \frac{n_0}{n}\int_0^\infty \frac{p\,dp}{2\pi\hbar^2 n} J_0\left(\frac{pr}{\hbar}\right)\left(\frac{1}{\sqrt{1 + 4mgn/p^2}} - 1\right), \tag{14}$$

where $J_0(x)$ is the zero-order Bessel function of the first kind, and the term $\sim (n - n_0)^2/n^2$ was neglected. Alternatively, the pair distribution function $g_2(r)$ can be obtained from the Fourier transform of the static structure factor $S(p)$. In the single-mode approximation $S(p)$ is related to the Bogoliubov spectrum $E_p$ via the Feynman formula $S(p) = p^2/(2mE_p)$ [52, 53]. The resulting expression differs from Eq. (14) by the absence of the $n_0/n$ factor. This discrepancy arises because, within the Bogoliubov theory up to the considered order of accuracy, there is no

distinction between the condensed density $n_0$ and the total density $n$. Therefore, the presence of $n_0$ is an artifact of the Bogoliubov approximation and $n_0$ should be substituted by $n$.

Expression (14) for the pair correlation function is valid in the long-wavelength limit, i.e. for all $r \gg a$. One of the distinct regimes within this limit is the hydrodynamic range of small momenta ($r \gg \xi$) where $g_2(r) \approx 1 - \xi/(2\pi\sqrt{2}nr^3)$. This range lies outside of the universal regime and cannot be matched with the two-body scattering problem. We are interested in the other limit, i.e. the intermediate regime $r \ll \xi$ in which the integration yields

$$\frac{g_2(r)}{n^2} = 1 - \frac{mg}{\pi\hbar^2} \ln\left(\frac{8}{e^{2\gamma_E+1}}\frac{\xi^2}{r^2}\right) + \mathcal{O}\left(\frac{r^2}{\xi^2}\right). \tag{15}$$

The details of this calculation are provided in the Appendix B. Comparing to Eq. (10), we find

$$\varkappa = \frac{8}{e^{2\gamma_E+1}} \approx 0.927752. \tag{16}$$

## 2.3 The Tan's contact and universal relation between the chemical potential and the potential energy density

Matching Eqs. (15) and (10) for $g_2(r)$ which are both valid in the range $a \ll r \ll \xi$ allows us to define the Tan's contact $\mathcal{C}$. We use the beyond-mean-field equation of state of a 2D Bose gas [54]

$$n = \frac{m\mu}{4\pi\hbar^2} \ln\frac{4\hbar^2}{m\mu a^2 e^{2\gamma_E+1}}\left(1 + \mathcal{O}\left(\ln^{-2}\frac{\hbar^2\varkappa}{2m\mu a^2}\right)\right), \tag{17}$$

where $\mu$ is the chemical potential. In order to get the higher-order relation between the coupling constant $g$ and the $s$-wave scattering length $a$, equivalent to renormalization, we relate $g$ to the inverse adiabatic compressibility according to

$$g = \frac{\partial\mu}{\partial n} \approx \frac{4\pi\hbar^2}{m\ln(\hbar^2\varkappa/2m\mu a^2)} + \mathcal{O}\left(\ln^{-2}\frac{\hbar^2\varkappa}{2m\mu a^2}\right), \tag{18}$$

where the last term in the parentheses is a correction, since $a \ll \xi$. In terms of the healing length $\xi^2 = \hbar^2/(2m\mu)$ therefore

$$g \approx \frac{4\pi\hbar^2}{m\ln(\varkappa\xi^2/a^2)} + \mathcal{O}\left(\ln^{-2}\frac{\varkappa\xi^2}{a^2}\right). \tag{19}$$

Substitution to Eq. (15) yields

$$\frac{g_2(r)}{n^2} = 1 - 2\frac{\ln(\varkappa\xi^2/r^2)}{\ln(\varkappa\xi^2/a^2)} + \mathcal{O}\left(\ln^{-2}\frac{\varkappa\xi^2}{a^2}\right). \tag{20}$$

Comparing to Eq. (10), we find the value of Tan's contact of the 2D Bose gas with power-law interactions:

$$\mathcal{C} = \frac{(4\pi n)^2}{\ln^2(\varkappa\xi^2/a^2)} + \mathcal{O}\left(\ln^{-4}\frac{\varkappa\xi^2}{a^2}\right). \tag{21}$$

Thus $\mathcal{C}$ is dependent on the interaction exponent $\alpha$ via the scattering length $a$ [see Eq. (5)].

Finally, expressing the chemical potential $\mu$ from the equation of state (17)

$$\mu = \frac{4\pi\hbar^2 n}{m\ln(\varkappa\xi^2/a^2)} + \mathcal{O}\left(\ln^{-3}\frac{\varkappa\xi^2}{a^2}\right) \tag{22}$$

and combining with Eq. (21), we get from Eq. (9) the universal relation for the chemical potential and potential energy per particle:

$$\frac{U}{N} = \frac{m\mu^2}{4\pi\hbar^2 n(\alpha-2)} \tag{23}$$

with a relative error of $\mathcal{O}[\ln^{-2}(\varkappa\xi^2/a^2)]$ at $a \ll \xi$. We note that Eq. (23) does not depend on the interaction amplitude $Q$, and that the resulting potential energy density (9) does not explicitly depend on the particle density $n$: $u_{2D} \propto \mu^2/(\alpha-2)$.

It is important to note, that in our approach to the calculation of the contact $\mathcal{C}$ and derivation of the universal relations (9) and (23), we do not rely on the adiabatic sweep theorem [10, 11, 46] (i.e. did not calculate the variation of the system's energy with the change of the $s$-wave scattering length). Moreover, the result obtained for $\mathcal{C}$ is more accurate compared to the one obtained in the universal regime (see e.g. [48] for dipoles): in Eq. (21), as the logarithm in the denominator contains $\xi^2$ that is, in turn, also logarithmically dependent on $na^2$, i.e. $\mathcal{C} \sim (n/\ln[\ln(1/na^2)/na^2])^2 + \dots$ [see Eqs. (17) and (19)], that is the coefficient under the logarithm and the subleading term are derived. This order of accuracy cannot be achieved when a finite-range interaction potential is approximated by the contact pseudopotential (see the detailed discussion of the universal regime in Ref. [55]).

# 3 Three dimensions

For three-dimensional Bose gas at $T = 0$ interacting via the potential (1), we will develop the theory of the Tan's contact and obtain universal relations in analogy with the 2D case. differently from the two-dimensional case, our considerations here do not include dipolar interactions $\alpha = 3$, since the matching of the two-body scattering problem at low energies with the many-body problem in macroscopic limit is valid only for $\alpha > 3$ (i.e. quadrupole interaction, van der Waals potentials, etc.).

## 3.1 Two-body scattering problem and universal relation for potential energy

We consider the 3D Schrödinger equation of the relative two-body motion at zero energy:

$$\left[-\frac{\hbar^2}{m}\frac{1}{r^2}\frac{\partial}{\partial r}\left(r^2\frac{\partial}{\partial r}\right) + \frac{Q}{r^\alpha}\right]\psi(r) = 0 \tag{24}$$

with the boundary conditions $\psi(0) = 0$ and $\psi(\infty) = 1$. Using dimensionless units, $\rho = r/r_0$ with $r_0 = (mQ/\hbar^2)^\nu$, $\nu = 1/(\alpha-2)$ and $z = 2\nu/\rho^{1/(2\nu)}$ (same as before) brings Eq. (24) to the modified Bessel equation with the general solution

$$\psi(\rho) = \tilde{C}_1 \frac{(2\nu)^\nu}{\sqrt{\rho}} I_\nu\left[\frac{2\nu}{\rho^{1/(2\nu)}}\right] + \tilde{C}_2 \frac{(2\nu)^\nu}{\sqrt{\rho}} K_\nu\left[\frac{2\nu}{\rho^{1/(2\nu)}}\right],$$

where $I_\nu(z)$ and $K_\nu(z)$ are the modified Bessel functions of the $\nu$-th order. Similarly to the 2D case, the divergence at $\rho \to 0$ is avoided by setting $\tilde{C}_1 = 0$. The $s$-wave scattering length is obtained from the long-range expansion $\psi \sim x^\nu K_\nu(x)$, $K_\nu(x \to 0) \approx x^\nu 2^{-1-\nu}\Gamma(-\nu) + x^{-\nu}2^{-1+\nu}\Gamma(\nu)$ by comparison with the asymptotic solution of the scattering problem $\lim_{r\to\infty}\psi(r) = 1 - a/r$. The 3D $s$-wave scattering length can be calculated explicitly as

$$a = \nu^{2\nu}\frac{\Gamma(1-\nu)}{\Gamma(1+\nu)}r_0 = \frac{\Gamma[(\alpha-3)/(\alpha-2)]}{\Gamma[(\alpha-1)/(\alpha-2)]}\left(\frac{mQ}{\hbar^2(\alpha-2)^2}\right)^{1/(\alpha-2)}, \tag{25}$$

where $\Gamma(x)$ is the Gamma function. The solution to the two-body scattering problem (24) reads

$$\psi(r) = \frac{2\nu^{\nu}}{\Gamma(\nu)}\sqrt{\frac{r_0}{r}}K_{\nu}\left[2\nu\left(\frac{r_0}{r}\right)^{1/(2\nu)}\right] \quad \text{with} \quad \nu = \frac{1}{\alpha-2}. \tag{26}$$

We show characteristic examples of the resulting wave function in Fig. 1**b**. Notably, the ratio of the scattering length in three (25) and two (5) dimensions approaches unity,

$$\frac{a_{3D}}{a_{2D}} = \frac{\Gamma(1-\nu)}{e^{2\nu\gamma_E}\Gamma(1+\nu)} \to 1, \quad \alpha \to \infty. \tag{27}$$

The physical reason behind this is that steeper potentials (i.e., those with larger values of $\alpha$) resemble hard cores more closely. For hard-core potential, the $s$-wave scattering length $a$ is equal to the diameter of the hard core in any dimensionality. Thus the inset of Fig. 1**b** shows the behavior of the scattering length $a$ with the interaction exponent for both 2D and 3D cases and there is no need to introduce different notations.

As we have defined $\psi(r)$ to be dimensionless (cf. the regular asymptotic shape $1/a - 1/r$, see e.g. [46,47,56]), the relation between the pair correlation function at short distances and the wave function of the two-body scattering problem preserving the correct dimension is

$$\lim_{r \to 0} g_2(r) = \frac{\mathcal{C}}{16\pi^2 a^2}\psi^2(r). \tag{28}$$

For interaction potentials that are short-range in the strict sense (i.e. integrable at infinity), the limiting procedure (28) means the distance scales much smaller than the healing length $r \ll \xi$, where $\xi = 1/\sqrt{8\pi an}$ and $n = N/V$ is the 3D particle density, $V$ is the system volume. As the main contribution to the potential energy arises from the ultraviolet contribution, mean potential energy can be explicitly calculated relying on Eq. (28)

$$u_{\text{3D}} \equiv \frac{U}{V} = \frac{1}{2}\int \frac{Q}{r^{\alpha}}g_2(r)d^3\mathbf{r} = \frac{\hbar^2\mathcal{C}}{8\pi am(\alpha-2)}. \tag{29}$$

Combining the large-distance asymptotics of Eq. (26), i.e. $\psi(r) \approx 1 - a/r + \mathcal{O}[(a/r)^{1/\nu}]$, with the definition of the pair distribution function (28), we arrive at the expression directly suitable to the matching with the Bogoliubov expression in the range $a \ll r \ll \xi$:

$$\frac{g_2(r)}{n^2} = \frac{\mathcal{C}}{16\pi^2 a^2 n^2}\left[1 - \frac{2a}{r} + \mathcal{O}\left(\frac{a^2}{r^2}\right) + \mathcal{O}\left(\frac{a}{r}\right)^{1/\nu}\right] \tag{30}$$

which represents the desired connection between $g_2(r)$ and the Tan's contact $\mathcal{C}$ following from the scattering problem.

## 3.2 The Tan's contact and universal relation in 3D

For the many-body limit of the pair correlation function within the Bogoliubov theory, after integration over angles in 3D space, we get (cf. Eq. (14)):

$$\frac{g_2(r)}{n^2} = 1 - \frac{8a}{\pi r}\int_0^{\infty} \frac{t^2 dt}{16\pi anr^2}\frac{\sin t}{t}\left(1 - \frac{1}{\sqrt{1 + 16\pi anr^2/t^2}}\right), \tag{31}$$

where $t = pr/\hbar$ and we used the definition of the coupling constant in 3D $g = 4\pi\hbar^2 a/m$ (see e.g. [56]). Given that $16\pi anr^2 = 2(r/\xi)^2 \ll 1$ in the range where we aim to match with the

result of the scattering problem (30), the integration yields (the details of the calculation are presented in Appendix B):

$$\frac{g_2(r)}{n^2} = 1 - \frac{2a}{r} + \frac{64}{3\sqrt{\pi}}\sqrt{na^3} + \mathcal{O}(a^2). \tag{32}$$

Comparing to Eq. (30), we obtain an explicit expression for the Tan's contact

$$\mathcal{C} = (4\pi na)^2 \left( 1 + \frac{64}{3\sqrt{\pi}}\sqrt{na^3} + \cdots \right). \tag{33}$$

The obtained expression for the two-body contact for power-law interacting Bose gas coincides with the prediction of the zero-temperature Bogoliubov theory for contact pseudopotential (see e.g. [57,58]) when including the Lee-Huang-Young correction [59] to the mean-field equation of state.

Some additional useful relations can be obtained by using the beyond-mean-field equation of state

$$\frac{E}{V} = \frac{2\pi\hbar^2 an^2}{m} \left( 1 + \frac{128}{15\sqrt{\pi}}\sqrt{na^3} + \mathcal{O}(na^3) \right) \tag{34}$$

to obtain the chemical potential by differentiating with respect to the density, according to

$$\mu = \frac{\partial}{\partial n}\frac{E}{V} = \frac{4\pi\hbar^2 an}{m} \left( 1 + \frac{32}{3\sqrt{\pi}}\sqrt{na^3} \right). \tag{35}$$

Combining Eqs. (29), (33) and (35), we get the universal connection between the chemical potential and potential energy per particle in three dimensions:

$$\frac{U}{N} = \frac{m\mu^2}{8\pi\hbar^2 an(\alpha - 2)} \tag{36}$$

with the relative error $\mathcal{O}(na^3)$ at $a \ll \xi$ (which is equivalent to the diluteness condition $na^3 \ll 1$). We note that Eq. (36) differs from its two-dimensional counterpart Eq. (23) by the dimensional factor of $1/(2a)$. Similarly, in the 2D case, the potential energy density $u_{3D}$ does not depend on $n$.

# 4   Summary

To conclude, we have derived universal relations for uniform quantum Bose gases with finite-range power-law interactions $V(r) \propto 1/r^\alpha$ in two and three dimensions. Importantly, our results do not rely on the adiabatic sweep theorem, differently from the original approach used by Shina Tan [10, 11]. As a result, the obtained expressions are applicable to potentials that cannot be substituted by zero-range ones. We explicitly define the zero-temperature Tan's contact parameter for any interaction exponent ($\alpha > 2$ in 2D and $\alpha > 3$ in 3D) without introducing the contact pseudopotential approximation. Instead, we match the pair correlation function found from the exact solution of the low-energy two-body scattering problem (valid for distances smaller than the healing length) and that from the many-body Bogoliubov approximation (valid in the long-wavelength limit). To demonstrate that the two-body scattering problem is relevant in the many-body picture up to the distances of the order of the healing length, we perform quantum Monte Carlo simulations.

In the two-dimensional case, we find a more accurate value for the contact $\mathcal{C}$ than that obtained from the adiabatic sweep theorem (for dipoles, see Ref. [48]). For three dimensions,

we show that the contact coincides with that for gases with zero-range interactions, however, in our case the dependence on $\alpha$ is explicitly included via the scattering length.

For the case of finite-range interactions considered here, we define the potential energy density in terms of the contact, and therefore show the validity of thermodynamic relations. Including beyond-mean-field corrections to the equation of state (see Mora and Castin [54] for 2D case and Lee, Huang and Young [59] for three dimensions) allows one to accurately connect the chemical potential with the potential energy per particle. Results obtained in this work including the new universal relations are summarized in Table 1.

| Two dimensions | Three dimensions |
|:---:|:---:|
| $\alpha > 2$ | $\alpha > 3$ |
| $a = e^{2\gamma_E/(\alpha-2)}\left[\dfrac{mQ}{\hbar^2(\alpha-2)^2}\right]^{\frac{1}{\alpha-2}}$ | $a = \dfrac{\Gamma[(\alpha-3)/(\alpha-2)]}{\Gamma[(\alpha-1)/(\alpha-2)]}\left[\dfrac{mQ}{\hbar^2(\alpha-2)^2}\right]^{\frac{1}{\alpha-2}}$ |
| $\mathcal{C} = \dfrac{(4\pi n)^2}{\ln^2\left[\dfrac{\ln\{\ln(\ldots)/(\pi e^{2\gamma_E+1}na^2)\}}{\pi e^{2\gamma_E+1}na^2}\right]}$ | $\mathcal{C} = (4\pi na)^2\left(1 + \dfrac{64}{3\sqrt{\pi}}\sqrt{na^3}\right)$ |
| $\dfrac{U}{N} = \dfrac{m\mu^2}{4\pi\hbar^2 n(\alpha-2)}$ | $\dfrac{U}{N} = \dfrac{m\mu^2}{8\pi\hbar^2 an(\alpha-2)}$ |
| $u_{\text{2D}} = \dfrac{m}{4\pi\hbar^2}\dfrac{\mu^2}{(\alpha-2)}$ | $u_{\text{3D}} = \dfrac{m}{8\pi\hbar^2}\dfrac{\mu^2}{a(\alpha-2)}$ |

Table 1: The scattering length $a$, the Tan's contact $\mathcal{C}$, and the relations for potential energy and chemical potential of the zero-temperature Bose gas with power-law interactions, for two and three dimensions. $\alpha$ denotes the interaction exponent, $Q$ is the amplitude of the potential.

## Acknowledgements

**Funding information**    This work has been supported by the NRNU MEPhI Priority 2030 Program and by the Spanish Ministry of Science and Innovation (MCIN/AEI/10.13039/501100011033, grant PID2020-113565GB-C21), by the Spanish Ministry of University (grant FPU No. FPU22/03376 funded by MICIU/AEI/10.13039/501100011033), and by the Generalitat de Catalunya (grant 2021 SGR 01411). I.L.K. acknowledges the support of Project No. FFUU-2024-0003.

## A  Finding the range of the applicability of the two-body scattering in the many-body problem

We perform variational Monte Carlo simulations for the $V(r) \propto 1/r^6$ interaction potential considered as an example to determine the maximal distances at which the two-body problem remains relevant within a many-body context. To do so, we consider the many-body trial wave function in a pair product form of Jastrow terms, constructed from the two-body scattering solution at short distances and a hydrodynamic asymptotic behavior [60] at large distances. The trial wave function is parameterized by two variables which are the scattering energy of the two-body problem and the matching distance $R_{\text{par}}$, where the two solutions are continuously

matched. These parameters are optimized by minimizing the variational energy in Monte Carlo simulations.

Figure 2 reports the optimal matching distance as a function of the energy per particle in the many-body system. We find that the matching distance scales linearly with the healing length over an extensive range of densities (with $nr_0^2$ ranging from $10^{-60}$ to $10^{-6}$). Although the considered densities might appear extremely low, a peculiarity of the two-dimensional geometry is that the coupling constant $g$ depends in a weak logarithmic way on density, so that the considered values of $g$ are not yet that small ($0.1 \lesssim g \lesssim 1$). To show that, we solve self-consistently the beyond-mean-field equation of state (17) of a 2D Bose gas [54]

$$\frac{mg}{\hbar^2} = \frac{4\pi}{\ln\left(\frac{4e^{-1-2\gamma}}{na^2}\frac{\hbar^2}{mg}\right)} \tag{37}$$

to extract the value of the coupling constant $g$ for each considered density $nr_0^2$.

This suggests that the scattering problem is relevant up to distances on the order of the healing length, rather than the mean interparticle distance as suggested in previous studies [48].

In order to demonstrate the excellent accuracy of the variational description, we compare the optimal variational energy (an upper bound) obtained using $R_{par}$ shown in Fig. 2 with the diffusion Monte Carlo energy (exact). We find a relative difference of only $10^{-4}$ for the coupling constant $g \approx 0.1\hbar^2/m$. Such a tiny difference confirms that the variational wave function is very close to the true ground state, indicating that its properties closely reflect those of the ground state. Hence, the variational wave function can be used to define the region of the applicability of the two-body scattering solution in the many-body context.

While the hydrodynamic asymptotic is expected to fail at distances of the order of the healing length and smaller, we argue that, in variational calculations, the energy contributions from the hydrodynamic part (i.e. corresponding to the small momenta, $k \to 0$ of the linear excitation spectrum $v_s k$) are smaller compared to those corresponding to the short distance (large momenta) behavior. Consequently, our variational procedure is primarily sensitive to the correct description of the short-range part, which is the focus of our validation in this study.

## B    Details of the integration

In order to calculate the integral in Eq. (14) in the limit $r \ll \xi$, we rewrite it using $\xi = \sqrt{\hbar^2/2mgn}$ as $g_2(r)/n^2 = 1 + I/(\pi n\xi^2 c^2)$, where

$$I = \int_0^\infty dx J_0(x)\left(\frac{x^2}{\sqrt{x^2+c^2}} - x\right) = \int_0^\infty dx J_0(x)\left(\sqrt{x^2+c^2}-x\right) - c^2 \int_0^\infty dx \frac{J_0(x)}{\sqrt{x^2+c^2}} \tag{38}$$

with $x = pr/\hbar$ and $c^2 = 2(r/\xi)^2$. Both integrals in the sum (38) will be considered as functions of $c$ in the limit $c \to 0$. The second is a table integral

$$\mathcal{I}_2(c) \equiv \int_0^\infty dx \frac{J_0(x)}{\sqrt{x^2+c^2}} = I_0\left(\frac{c}{2}\right)K_0\left(\frac{c}{2}\right),$$

while the first

$$\mathcal{I}_1(c) \equiv \int_0^\infty J_0(x)\left(\sqrt{x^2+c^2}-x\right)$$

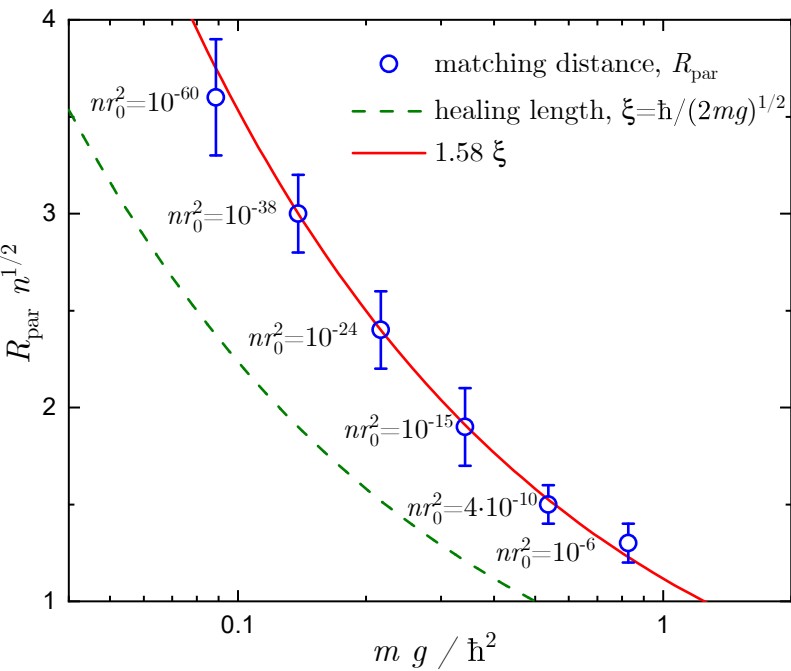

Figure 2: Maximal distance up to which the two-body solution applies in the many-body problem in the rarefied limit. Blue symbols, matching distance $R_{par}$ between the two-body scattering solution and the hydrodynamic long-range asymptotic as obtained from variational Monte Carlo optimization of the energy. Red solid line, the healing length $\xi$ scaled by a numerical constant $\mathbf{1.58}\xi$. Distances, shown on the vertical axis, are given in the units of the mean interparticle distance $n^{-1/2}$. Horizontal axis shows the 2D coupling constant.

can be found from the differential equation $\mathcal{I}'_1(c) = c\mathcal{I}_2(c)$ with the initial condition $\mathcal{I}_1(0) = 0$, so that

$$\mathcal{I}_1(c) = \int_0^c dt\, I_0\left(\frac{t}{2}\right) K_0\left(\frac{t}{2}\right) t = \frac{c^2}{2}\left[I_0\left(\frac{c}{2}\right)K_0\left(\frac{c}{2}\right) - I'_0\left(\frac{c}{2}\right)K'_0\left(\frac{c}{2}\right)\right].$$

Using the asymptotic behavior for the modified Bessel functions $I_0(c/2 \to 0) \approx 1 + c^2/16 + \mathcal{O}(c^3)$ and $K_0(c/2 \to 0) \approx \ln(4/ce^{\gamma_E}) + c^2/16\ln(4e/ce^{\gamma_E}) + \mathcal{O}(c^3)$, we get

$$\lim_{c\to 0}\frac{g_2(r)}{n^2} = 1 - \frac{1}{\pi n\xi^2}\ln\frac{4}{ce^{\gamma_E+1/2}} + \mathcal{O}(c^2) = 1 - \frac{2mg}{\pi\hbar^2}\ln\frac{2\sqrt{2}}{e^{\gamma_E+1/2}}\frac{\xi}{r} + \mathcal{O}\left(\frac{r^2}{\xi^2}\right). \quad (39)$$

The integral in Eq. (31) is calculated likewise. Introducing the notation $16\pi anr^2 = \eta^2$, or $\eta = \sqrt{2}r/\xi \ll 1$, we get

$$\frac{g_2(r)}{n^2} = 1 - \frac{8a}{\pi r}\left[\int_0^\infty \frac{\sin t}{\sqrt{t^2+\eta^2}}dt - \frac{1}{\eta^2}\int_0^\infty (\sqrt{t^2+\eta^2} - t)\sin t\, dt\right]. \quad (40)$$

The first integral in the square brackets

$$\mathcal{I}_3(\eta) \equiv \int_0^\infty \frac{\sin t}{\sqrt{t^2+\eta^2}}dt = \frac{\pi}{2}\left[I_0(\eta) - L_0(\eta)\right],$$

where $I_k(\eta)$ and $L_k(\eta)$ are the modified Bessel and Struve functions of $k$-th order, respectively. The second integral in the brackets

$$\mathcal{I}_4(\eta) \equiv \int\limits_0^\infty \sin t (\sqrt{t^2 + \eta^2} - t) dt$$

satisfies $\mathcal{I}_4'(\eta) = \eta \mathcal{I}_3(\eta)$ with the boundary condition $\mathcal{I}_4(0) = 0$, which yields

$$\mathcal{I}_4(\eta) = \eta \frac{\pi}{2} \big[ I_1(\eta) + L_1(\eta) \big].$$

As a result,

$$\frac{g_2(r)}{n^2} = 1 - \frac{a}{r} \left[ \frac{1}{\eta} L_1(\eta) - \frac{1}{\eta} I_1(\eta) + I_0(\eta) - L_0(\eta) \right] \tag{41}$$

with the asymptotic at $\eta \to 0$ given by Eq. (32). Note that in the asymptotic expression obtained from (41), first one needs to take the limit $a \to 0$ and only then $r \to \infty$.

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
