# Peer review of "Universal relations in Bose gases with power-law interactions in two and three dimensions"

_SciPost Physics_

## Round 1 · Referee Report · Anonymous (Referee 1) · 2024-10-15

Report

Brief overall assessment:

After a quite careful reading of the manuscript and of the existing literature, I am sorry to conclude that most of the content of the manuscript is either not novel, or likely to be incorrect, or merely a nice consistency check of Bogoliubov theory. Therefore, I cannot recommend publication in SciPost Physics or in SciPost Physics Core, even if major revisions were done.

Detailed report:

-The relation between potential energy and contact in 2D, Eq.(9), already appeared in Eqs.(C9) and (C10) of the reference

J. Hofmann and W. Zwerger, Phys. Rev. Res. 3, 013088 (2021)
(Ref. [48] in the manuscript)

About that reference, the authors write:

"Since we consider the amplitude $Q$ of the interaction potential fixed, no other contact value arises in our description from the change of the energy with respect to variation of $Q$ (cf. Ref. [48] where the variation of the dipole length led to the appearance of a specific relation with a dipolar contact)"

This suggests that the authors misunderstood the reason for the appearance of a dipolar contact in Ref. [48]. The actual reason is that, while the present manuscript only considers a pure power-law interaction potential (i.e. a power-law for all $r$) , Ref. [48] also considers the more general situation where the potential deviates from the power-law at small $r$ (this is relevant for dipolar atomic gases near a Feshbach resonance).

The authors also write:

"As pointed out in Ref. [48], the effective interaction of dipoles when truncated to motion in the plane needs to contain the short-distance cutoff that ensures the existence of a proper zero-range scaling limit. Our description does not require such a cutoff as we take the finite range of interactions into account."

This also suggests that the authors misunderstood something about Ref.[48]. The pure dipolar case which is discussed in various parts of Ref.[48] is exactly the same hamiltonian than the one considered in the present manuscript (and this hamiltonian does not contain any cutoff).

-One of the main claims of the authors is their expression for the contact in 2D, Eq.(21), about which they write:

"the result obtained for $\mathcal{C}$ is more accurate compared to the one obtained in the universal regime (see e.g. [48] for dipoles): in Eq. (21), as the logarithm in the denominator contains $\xi^2$ that is, in turn, also logarithmically dependent on $na^2$ , i.e. $\mathcal{C} \sim (n/ \ln[\ln(1/na^2 )/na^2 ])^2 + ...$ [see Eqs. (17) and (19)], that is the coefficient under the logarithm and the subleading term are derived. This order of accuracy cannot be achieved when a finite-range interaction potential is approximated by the contact pseudopotential"

I do not believe that the last sentence (" This order of accuracy cannot be achieved when a finite-range interaction potential is approximated by the contact pseudopotential") can be correct, because Eq.(10) can be obtained by replacing $\psi(r)$ with $\ln(r/a)$ in Eq.(7), which means that Eq.(7), and by extension the entire reasoning leading to Eq.(21), is by no means specific to power-law interactions.

Furthermore, I see no reason to trust the validity of Eq.(21), given how it was derived:

*In Eq.(18) the authors use a relation, $g = \frac{\partial \mu}{\partial n}$, without providing any justification. This relation is very strange, since $g$ is the bare coupling constant appearing in the hamiltonian whereas $\frac{\partial \mu}{\partial n}$ is a physical quantity which depends on the density. $g$ should not depend on the density. Instead, it should depend on an ultraviolet cutoff, which is needed to regularize the Dirac $\delta$ interaction in 2D. I am aware that $\mu = g n$ at the mean-field level, but this is a priori not sufficient here since the goal is to make a statement about higher-order corrections.

*It is also rather puzzling that it is important in the reasoning of the authors to fix the constant $\kappa$ to a particular value, while this parameter was initially arbitrary.

Finally I see no reason to doubt the validity of the expression for $\mathcal{C}$ obtained in Ref.[48].

-The expression for the scattering length in 2D, Eq.(5), already appeared in Appendix C of Ref.[48] for any value of the exponent.

-The expressions Eqs. (7) and (28) of $g_2$ in terms of $\mathcal{C} \psi^2(r)$ , with $\psi$ the zero-energy two-body scattering wavefunction, together with the statement that the condition $r \ll \xi$ (with $\xi$ the healing length) is sufficient for this expression to hold for weakly interacting bosons, already appeared for the 2D case in

A. Y. Cherny, J. Phys. A 55, 155004 (2022)

and for the 3D case in

A. Y. Cherny, Phys. Rev. A 104, 043304 (2021)

-For the expression of $g_2$ from Bogoliubov theory in 3D, Eq.(32), the first two terms, together with the condition $r \ll \xi$, were already obtained from Bololiubov theory in

M. Holzmann and Y. Castin, Eur. Phys. J. D 7, 425 (1999)

and they were also found in the Lee-Huang-Yang paper [Phys.
Rev. 106, 1135 (1957)].

-As far as I know, the third term in Eq.(32) (the $\sqrt{n a^3}$ term) does not appear in the literature, and the authors carried out a nice consistency check by showing that the resulting beyond-mean-field term in the expression of the contact, i.e. the $\sqrt{n a^3}$ term in Eq.(33), agrees with the result obtained from the adiabatic theorem by taking the derivative of the Lee-Huang-Yang correction to the energy.

-It should be clearly stated that the interactions are assumed to be repulsive (otherwise a short-distance cutoff or regularization would be needed for the scattering length and all other quantities to be well-defined). This reduces the experimental relevance, especially for the 3D case, since many of the claimed applications (e.g. van der Waals interactions) are usually in the attractive case.

-It should be clearly stated that the entire paper is restricted to the weakly interacting regime ($n a^d \ll 1$ with $d$ the space dimension).

-In Appendix A, variational Monte Carlo is used to provide evidence for the statement that Eq.(2) is valid for $r \ll \xi$. This is a rather nice observation, although this validity range was already stated and derived from Bogoliubov theory in 3D in the literature (see above), and although the evidence is rather indirect, since it comes from the optimal value of a parameter of the variational wavefunction. Moreover this wavefunction is not given explicitly so that this parameter remains quite obscure.

Recommendation

Reject

---

## Round 1 · Referee Report · Anonymous (Referee 2) · 2024-10-22

Strengths

1- The authors have found beyond mean-field corrections for the Tan's contact in 2D bosonic systems with power law interactions. 2- The approach that does not rely on the adiabatic sweep theorem that is valid only for zero-range potentials, but on the match of the pair correlation function evaluated at small distances and that evaluated from the Bogoliubov approximation

Weaknesses

1- The manuscript is not clear/ accurately written. Some examples: (i) the sentence in the introduction concerning short-range and long-range potentials needs to be clarified. I think that its final "...to differentiate them from short-range potentials" should read " ...to differentiate them from finite-range potentials"
(ii) In the two dimensions section the authors say that the scattering solution(6) and the scattering length (5) are shown in Figs. 1a and b (inset) BUT Fig1b refers to the three dimensional case (iii) The coupling constant g appear in the definition of the healing length, before Eq. 8, without being defined! The first expression for g is given in Eq. 18. (iv) Eq 7 and Eq. 28 have been derived for the case of finite-range potentials that is not the case treated in this work. The authors should explain why these relations should still be valid (v) etc... 2- Illustrative figures are missing (the scattering length in 2D, direct calculation of the contact..) 3- It is not clear that the 2D result obtained is more accurate compared to that obtained in the universal regime without a direct calculation of the contact that shows this. 4- It is not clear if the contact as defined in this work still define the amplitude of the momentum distribution tails. If yes, it would be important to compare numerically the contact as extracted from the momentum distribution with respect the formula found in this work.

Report

The actual version of this work is not suitable to be published in SciPost: (i) it is not clearly/accurately written so that it is not obvious to evaluate the validity of their approach and (ii) the main results are not validated by some numerical calculations (the authors, being expert in MonteCarlo simulations, can do it)

Requested changes

1- Please make clearer the discussions on short-/long-range potentials in the introductions 2 - Explain in the 2D case the relation between your results and the new dipolar contact of Hofmann and Zwerger 3- Add an inset or a figure for the 2-D scattering length, and adjust the reference to <fig. 1 in the text 4- Define the coupling constant g when it appears for the first time 5 - Eq. 10 is valid only in the limit of a->0 (that can be reached or if Q->0, or if alpha>>>2). This means that this is the validity of the contact expression in 2D (Eq. 21) as well as of the potential energy per particle (Eq. 23) rely on the same assumptions. Thus your claim that Eq. 23 does not depend on the amplitude Q it is not completely correct. Please discuss more accurately the validity of the results obtained in this section. 6- Please discuss more in details why Eqs (6) and (28) should be valid also in the long-range potential limit that you are considering in this work (alpha>D) 7- It would be important to have a numerical calculation of the g2 in 2D at least in order to validate the analytical expression found. A figure (or more figures) showing this comparison for different values of Q and alpha is actually missing. 8- A discussion concerning the momentum distribution and the contact for such long-range potentials is missing

Recommendation

Ask for major revision

---

## Editorial Decision

in_refereeing